# Air-CAD: Edge-Assisted Multi-Drone Network for Real-time Crowd Anomaly Detection

Paper #299

## ABSTRACT

Drones connected via the web are increasingly being used for crowd anomaly detection (CAD). Existing solutions, however, face many challenges, such as low accuracy and high latency due to drones' dynamic shooting distances and angles as well as limited computing and networking capabilities. In this paper, we propose Air-CAD, an edge-assisted multi-drone network that uses air-ground cooperation to achieve fast and accurate CAD. Air-CAD consists of two stages: person detection and multi-feature analysis. To improve CAD accuracy, Air-CAD dynamically adjusts the inference of person detection model based on drones' shooting distances and assigns appropriate feature analysis tasks to drones shooting at variable angles. To achieve fast CAD, edge devices connected to drones are deployed to offload assigned feature analysis tasks from drones. Air-CAD schedules the connection between each drone and edge to accelerate processing based on drone's assigned task and the computing/network resources of the edge device. To validate the performance of Air-CAD, we generate a new simulated human stampede dataset captured from various drone-view recordings. We deploy and evaluate Air-CAD in both simulation and real-world testbed. Experimental results show that Air-CAD achieves 95.33% AUROC and real-time inference latency within 0.47 seconds.

## KEYWORDS

Multi-drone network, Mobile edge computing, Anomaly detection

## 1 INTRODUCTION

As urban populations continue to grow, the risk of crowd disasters in cities like stampedes is increasing. Combined with Web of Things (WoT) technologies, crowd disasters can be prevented by using networked surveillance cameras to detect crowd disaster anomalies, such as the high density and the abnormal flow speed of the crowd [7]. However, these fixed cameras always have limited field-of-views (FoVs) and poor mobility for monitoring real-time changing crowds [4]. Networked drones, which offer wider FoVs and mobility, can detect anomalies and report them more promptly via the web than fixed cameras [10, 25, 35].

Recently, feature-based crowd anomaly detection (CAD) has gained popularity, comprising two stages: person detection and feature analysis [34]. While these techniques are highly accurate in surveillance camera views, they are less accurate in drone views, due to a variety of factors caused by high-flying drone views, e.g., the small scale, distortion, and blur of images [17].

In response to the inaccuracy challenges posed by drone views, some solutions have been proposed. In the person detection stage, Zhu et al. propose a drone-view object detector that uses deeper Transformer Prediction Heads (TPH) to improve the accuracy [37]. In the feature analysis stage, Reiss et al. design a multi-feature analysis method, which fuses the results from different types of features to achieve accurate CAD [28].

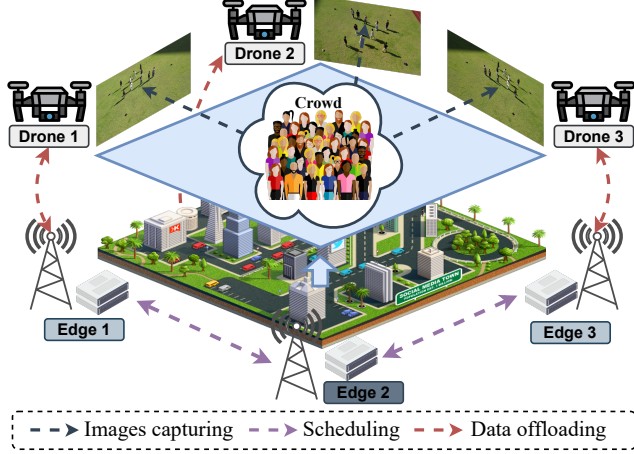

**Figure 1: Air-CAD uses multiple drones and edge devices to detect crowd disaster anomalies in urban scenarios.**

However, existing improvements do not perform well when deployed on real-world drones due to the *flight conditions* of drones, including the far shooting distances and the large shooting angles (collectively referred to as *shooting parameters*), as well as the limited computing/network resources [10]. Under different shooting parameters, the deeper detector may not always be superior, and the performance of feature analysis also varies severely. Limited hardware resources of drones also restrict the real-time execution of the deeper detector and sequential multi-feature analysis. These challenges are analyzed in detail in our motivational studies (§2). In addition, our goal is to detect anomalies that lead to dangerous crowd disasters, which are not included in current datasets.

In this paper, we present Air-CAD, an edge-assisted multi-drone network, which provides an air-ground cooperative WoT system for monitoring and reporting risks of crowd disasters. For instance, as shown in Figure 1, each drone captures images of the crowd and performs person detection at a shooting instant. Images from a drone are offloaded to a designated edge device, which performs an assigned type of feature analysis task to detect anomalies.

To achieve the above high-level design goals of Air-CAD, the following challenges need to be addressed: i) How to achieve fast and accurate person detection on drones with various shooting parameters? ii) How to assign suitable feature analysis tasks and edge devices for heterogeneous drones? iii) How to evaluate the performance of detecting crowd disaster anomaly without the datasets containing crowd disasters?

To tackle the aforementioned challenges, we design a two-stage pipeline for Air-CAD to achieve fast and accurate CAD on drones. In the person detection stage, we introduce a *zoom detector* that can detect people on multiple drones with various shooting distances. By perceiving the shooting distances, the zoom detector adjusts the depth of the model inference and focuses on the key channels

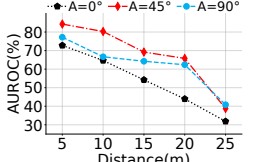 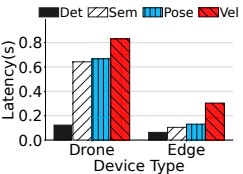 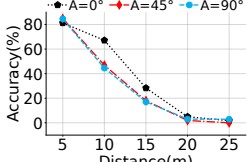 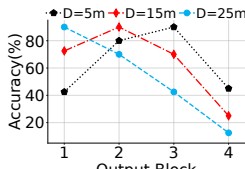 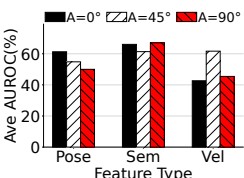

(a) Impact of shooting param-
eters on overall AUROC.

(b) Impact of computing re-
sources on the latency.

(a) Impact of shooting parame-
ters on detection accuracy.

(b) Impact of different depth
blocks' output on accuracy.

(c) Impact of shooting angles
on different feature analysis.

**Figure 2: Performance of general CAD.**

**Figure 3: Performance of two stages in general CAD.**

of the feature map. In the feature analysis stage, we propose a *feature scheduler* that improves multi-feature analysis accuracy in multi-drone views and reduces the system's processing latency. The feature scheduler trained by our model-assisted reinforcement learning could assign an optimal feature analysis task for the drone and select a suitable edge device to execute this task based on the shooting parameters and the computing/network resources.

To evaluate Air-CAD, we generate a new dataset for detecting crowd disaster anomalies, which records human stampedes from various multi-drone views. We deploy and evaluate Air-CAD in both simulation and real-world WoT drones. The evaluation results show that Air-CAD achieves 95.33% Area Under the Receiver Operating Characteristic curve (AUROC) and achieves real-time inference latency within 0.47 seconds, 14.25% higher and 1.72 times faster than the optimal baseline.

In summary, our key contributions are as follows:

- An edge-assisted multi-drone network for accurate and real-time CAD under flight conditions of real drones.
- A zoom detector to achieve fast and accurate person detection on real drones, which could adjust the depth of model inference and focus the key channel of the feature map based on drones' shooting distances.
- A feature scheduler for multi-feature analysis anomaly detection, which can allocate the most suitable tasks and edge devices for the drones based on drones' shooting parameters and edges' computing/network resources.
- An image dataset captured from multi-drone views, which can be used to evaluate crowd disaster anomaly detection across various drone views.

## 2 MOTIVATIONAL STUDIES

In this section, we analyze the impacts of a drone's flight conditions on the performance of general CAD and its two stages.

### 2.1 Implementation and Data

**Implementation.** We use two types of devices to simulate different computing capabilities: i) An NVIDIA Jetson Xavier NX as drone's onboard device. ii) A PC as the edge device, powered by an Intel Xeno E5-2687W v4 and an NVIDIA RTX 2080Ti. The CAD pipeline consists of two stages: i) TPH-Yolov5 [37] for person detection, which contains 4 blocks from shallow to deep that can output results. ii) AI-VAD [28] for feature analysis, which is a multi-feature analysis algorithm taking into account semantics, velocity, and pose.

**Data.** We create a new large-scale dataset of crowd disasters, which captures aerial images from multi-drone views with different shooting distances and angles, detailed in §5.

### 2.2 Impact on Performance of General CAD

**AUROC.** Figure 2a shows the AUROC of the general CAD for detecting crowd disaster anomalies. The AUROC of the general CAD is acceptable at close shooting distances (e.g., 5 meters) but decreases as the shooting distance increases. Meanwhile, the AUROC varies under three different shooting angles at 0°, 45°, and 90°.

**Latency.** Figure 2b shows the computation latency of the general CAD on the drone and edge device. As compared to the edge device, the latency of each stage of the pipeline increases severely on the drone device. Furthermore, there are significant differences in the computation latency of different feature analysis tasks.

**Insight #1.** *General CAD can be used to detect the crowd disaster anomalies, but the performance is restricted by real drone's flight conditions, including shooting parameters and hardware resources.*

### 2.3 Impact on Person Detection Performance

**Detection Accuracy.** Figure 3a shows the impact of shooting parameters on the person detection accuracy. As the shooting distance increases, the accuracy decreases significantly. In contrast, shooting angles have little impact on detection accuracy.

Furthermore, we output the detection results from blocks with different depths in the detector to analyze the variations in the accuracy, shown in Figure 3b. At a close shooting distance (e.g., 5 meters), a deeper block (e.g., block 3) would be necessary to achieve high accuracy, while at a far shooting distance (e.g., 25 meters), a shallower block (e.g., block 1) would suffice.

**Insight #2.** *The person detection accuracy on drones can significantly benefit from shooting distances, provided that the depth of model inference could be correctly determined based on shooting distances.*

### 2.4 Impact on Feature Analysis Performance

**AUROC.** Figure 3 shows the impact of shooting angles on the AUROC of feature analysis tasks. Due to the fact that different feature analysis tasks perform differently under the same shooting angle, one simple relationship cannot be drawn between each feature and its shooting parameters.

**Insight #3.** *It is also crucial to choose the optimal feature analysis task for drones with different shooting angles in order to enhance the performance of CAD based on multi-feature analysis.*

### 2.5 Summary and Motivation

The results above indicate that flight conditions significantly affect the performance of general CAD, which can be used to inspire the stages in the pipeline for better performance. Taking these factors into consideration, we design Air-CAD in the following ways: i)

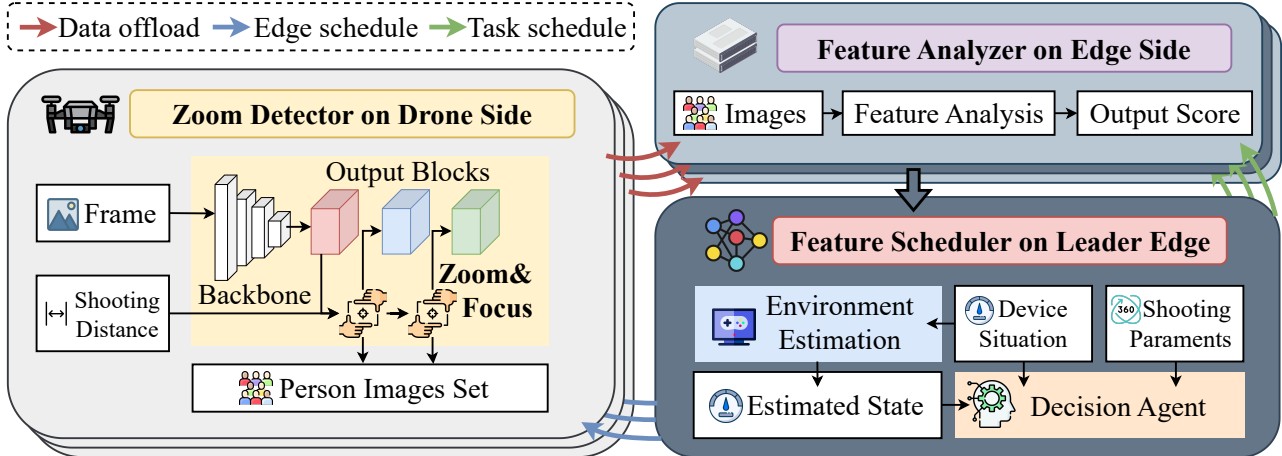

**Figure 4: Operational flow of Air-CAD in a nutshell.**

using networked multiple drones to obtain images with various shooting parameters, while assisting them with edge devices that provide more powerful hardware resources. ii) perceiving the flight conditions of drones to achieve fast and accurate CAD.

## 3    AIR-CAD OVERVIEW

**Goal.** For better CAD performance, we intend to design a multi-drone network that is able to perceive flight conditions.

**Deployment Scenario.** Figure 1 shows the deployment scenario of Air-CAD. The maneuverability of drones allows us to deploy Air-CAD to most surveillance scenarios, including obstructed urban areas. This edge-assisted multi-drone network can detect crowd disaster anomalies in real-time in the scene, which consists of:

①  a group of **Drones** at different flight conditions that could capture images and run lightweight models. Drones also have wireless communication capabilities that allow them to access the network to offload computing tasks to edge devices.

②  a group of **Edge Devices** connected with drones, which can efficiently run models but computing power is not unlimited. In Air-CAD, an edge device is selected as the **Leader Edge** (edge 2 in Figure 1), which collects information of drones and other edges (called **Cluster Edge**) to schedule feature analysis and data offloading.

**Overview.** The operational flow of Air-CAD is illustrated in Figure 4, which consists of the following components:

①  **Zoom Detector** on drone side (§4.1). Drones in Air-CAD use the zoom detector to detect people in the crowd. The zoom detector integrates a drone's shooting distance with its frame and outputs a person image set from this frame. This detector can adjust the depth of model inference and focus on the key channels of feature maps, thus achieving fast and accurate person detection.

②  **Feature Scheduler** on leader edge (§4.2). The leader edge uses the feature scheduler to schedule which edge device the drone should connect to (edge schedule in Figure 4) and which feature analysis task this edge should execute (task schedule in Figure 4). The scheduler takes into account the drones' shooting parameters and the edge devices' computing/network resources, and produces the matching pairs of drone-edge and edge-task. This process is powered by our model-assisted deep Q-Network, which includes a decision agent and an environment estimation agent.

③  **Feature Analyzer** on edge side (§4.3). According to the schedule from the leader edge, each drone sends the person image set to a corresponding edge device, which uses a feature analyzer to perform the assigned task. The analyzer inputs the person images set from the current frame, extracts the assigned features, and outputs the anomaly score of the current frame. Anomaly scores from all edges will be sent to the leader edge to obtain a fusion score, if it exceeds the threshold, it will be considered abnormal.

## 4    DETAILED DESIGN OF AIR-CAD

### 4.1    Zoom Detector

**Goal.** According to the motivational studies in §2, our goal is to determine the depth of model inference based on the shooting distances to achieve fast and accurate person detection.

**Design.** We develop a zoom detector for our goal. Compared to the original detector [37], we add Z&F blocks in our zoom detector to sense the shooting distances. Figure 5a illustrates the workflow of the zoom detector as follows:

①  **Infer.** Using the input image/feature map, the infer block creates a multi-channel feature map, which is then passed to the Z&F block for deciding whether to further inference.

②  **Zoom & Focus.** The Z&F block takes the feature map generated by the previous infer block and the drone's shooting distance as input, to perform the "zoom" and "focus" operations. The "focus" operation generates a focus weight to focus on the key channels of the feature map. The "zoom" operation generates a zoom signal to determine whether to continue inference. If this signal decides that further inference is needed, the Z&F block saves the current feature map and continues inferring, otherwise, it outputs the result.

③  **Output.** When the Z&F block makes a decision to output, the detector merges the previously saved feature maps to output a fused feature map. This feature map is processed by the prediction head to generate the people's bounding boxes, which can be used to crop out the person images set.

**Z&F Block.** The zoom detector perceives the shooting distance to enhance the accuracy and speed of detection through the proposed Z&F block. Figure 5b shows the structure of the Z&F block, which consists of the zoom and focus operations.

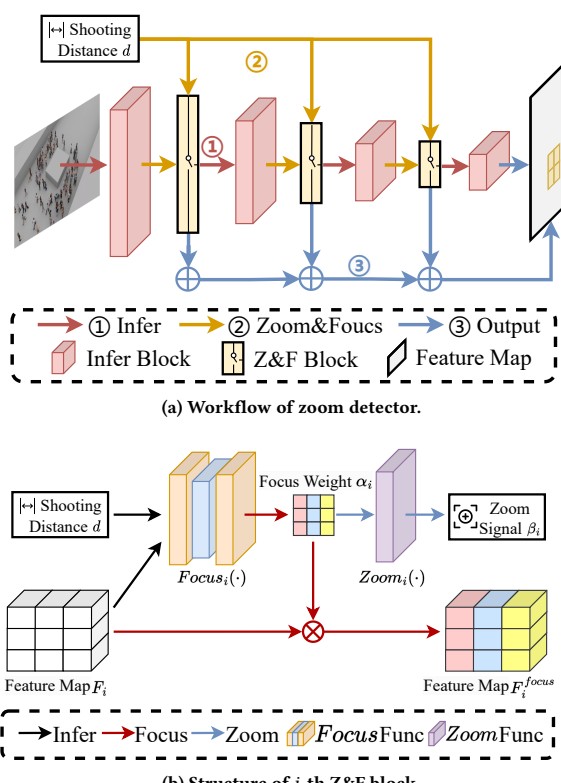

**(a) Workflow of zoom detector.**

**(b) Structure of $i$-th Z&F block.**

**Figure 5: Zoom detector for person detection.**

In focus operation, we generate a focus weight $\alpha_i \in \mathbb{R}^{1 \times C}$ based on feature map and shooting distance:

$$\alpha_i = Focus_i(Concat(AvgPool(F_i), \mathcal{N}(\frac{d}{d_{max}}, 1))), \quad (1)$$

where $F_i \in \mathbb{R}^{H \times W \times C}$ is the feature map outputted by infer block $i$. $H \times W$ is the spatial dimensions of $F_i$ and $C$ is the channels of $F_i$. $d$ and $d_{max}$ are the (maximum) shooting distances of the drone. $\mathcal{N}$ is the Gaussian distribution. The focus function $Focus_i(\cdot)$ is a non-linear function whose parameters can be updated. In this paper, the $Focus_i(\cdot)$ is composed of three fully-connected layers and two ReLU layers, which is inspired by squeeze-and-excitation block [18]. By using the focus weight $\alpha_i$, the focused feature map $F_i^{focus}$ is generated which is focused on key channels:

$$F_i^{focus} = Sigmod(\alpha_i) \times F_i. \quad (2)$$

In zoom operation, the $\alpha_i$ is used to generate a zoom signal to determine whether further inference is needed. The logarithm of the zoom signal is produced by the zoom function $Zoom_i(\cdot)$:

$$\beta_i = Zoom_i(\alpha_i), \quad (3)$$

where $\beta_i \in \mathbb{R}^{1 \times 2}$ represents the probability of continuing the inference after the infer block $i$. Same as the $Focus_i(\cdot)$, the $Zoom_i(\cdot)$ is also a non-linear function. In this paper, we use one fully-connected layer as the $Zoom_i(\cdot)$ to reduce inference latency. We generate the one-hot zoom signal $Z_i$ by the Gumbel Softmax function [36]:

$$Z_i^k = GumbelSoftmax(\beta_i^k | \beta_i). \quad (4)$$

Based on the $Z_i$, we can decide whether to further infer the detector. If the $Z_i$ is true, the detector saves $F_i^{focus}$ and transmits it to the infer block $i + 1$ for further inferring. Otherwise, the detector fuses all saved $F_i^{focus}$ to output the fused feature map, which is used to generate detection results.

**Inference Loss.** Our design goal is to obtain high accuracy with limited onboard computational resources. However, the detector trained using traditional loss functions tends to provide a sub-optimal solution (i.e., infer to the deepest infer block). We hope to minimize the number of inferred blocks while maintaining accuracy. Thus, we propose a multi-objective loss function to guide training:

$$\mathcal{L} = \mathcal{L}_{\mathcal{D}} + \lambda \mathcal{L}_{\mathcal{B}} = \mathcal{L}_{\mathcal{D}} + \lambda \frac{B - B_{min}}{B_{max} - B_{min}}, \quad (5)$$

where $\mathcal{L}_{\mathcal{D}}$ is the traditional detection loss and $\mathcal{L}_{\mathcal{B}}$ is the inference loss. $B$ is the number of blocks inferred in one detection. $B_{max}$ and $B_{min}$ are the maximum and minimum number of model blocks that can be used to generate results. The $\lambda$ is used to balance the detection accuracy and inference cost. With this loss function, we can achieve high accuracy while minimizing the inference depth of detectors, which results in efficient person detection.

## 4.2 Feature Scheduler

**Goal.** According to the motivational studies in §2, our goal is to decide: i) Which edge device should the drone offload data to? ii) Which feature analysis task should the edge device perform for processing the person image set of the connected drone?

**Design.** We propose a feature scheduler to achieve our goal. The feature scheduler can detect the shooting parameters of drones and the amount of computing/network resources of edge devices, and select the task and offloaded edge device in real-time. The scheduler is deployed on the leader edge, which consists of two components: a decision agent for scheduling and an environment agent for environment state estimation. In order to avoid frequent information exchange, the scheduler assigns a scheduling period (e.g., 10 seconds) to a series of tasks, expressed as $Q_a$. During a $Q_a$, the information of drones and edge devices is estimated by the environment agent. Figure 6 illustrates the workflow of the scheduler within a $Q_a$:

① **Synchronization.** At the start of the $Q_a$, the leader edge obtains all state information from the previous period, including the shooting parameters of drones and the computing/network resource of the cluster edges. Based on the obtained information, the environment agent updates itself to output accurate estimations of the environment. Then, the scheduler alternates steps ② and ③ to make decisions for all tasks in the $Q_a$.

② **Schedule Making.** For a task $Task_t$, the decision agent inputs state information $s_t$ and decides on the action $a_t$ (i.e. the edge device for offloading and the feature task it performs).

③ **State Estimation.** After ②, the environment agent inputs $s_t$ and $a_t$ to provide an estimated state $s'_{t+1}$ for the next task $Task_{t+1}$ and an estimated reward $r'_t$.

④ **Dissemination.** After all tasks in $Q_a$ are scheduled, the leader edge sends the schedule to all drones and cluster edges.

**Model-assisted Deep Q-Network.** To achieve the scheduler in leader edge, we propose a model-assisted Deep Q-Network, which aims to establish an optimal policy for edge device scheduling and

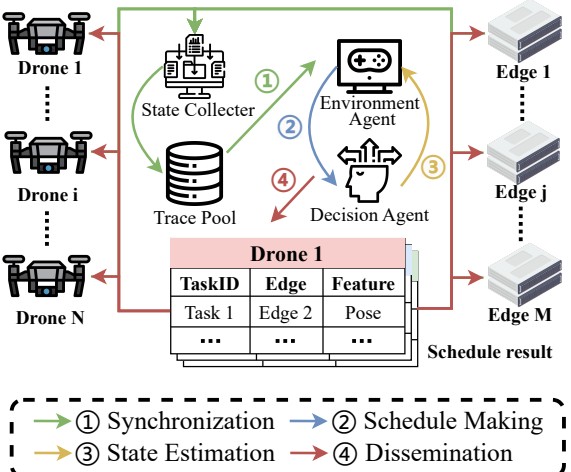

**Figure 6: Feature scheduler for multi-feature analysis.**

feature analysis task assignment [30]. Based on the basis DQN, model-assisted DQN learns two types of agents: i) a decision agent to learn the mapping relationship between state and action (i.e., $s_t \rightarrow a_t$) to make a sequence of decisions with the highest long-term gain, ii) an environment agent to learn the transition of environment state (i.e., $(s_t, a_t) \rightarrow (s_{t+1}, r_t)$) to simulate the environment.

**Decision Agent Learning.** The goal of the decision agent is to learn the mapping from $s_t$ to $a_t$. We consider $N$ drones $\mathbb{D} = \{D_1, \ldots, D_i, \ldots, D_N\}$, $M$ edge devices $\mathbb{E} = \{E_1, \ldots, E_j, \ldots, E_M\}$, and $L$ types of feature analysis tasks $\mathbb{T} = \{T_1, \ldots, T_k, \ldots, T_L\}$. In the $t$ decision, the decision agent observes the current state given by $s_t = \{s_t^d, s_t^e\}$. The state of drones $s_t^d$ is denoted as $s_t^d = \{Distance_t^i, Angle_t^i\}, i = \{1, \ldots, N\}$, where $Distance_t^i$ and $Angle_t^i$ represent the shoot distance and angle of drone $D_i$. The state of edge devices $s_t^e$ is denoted as $s_t^e = \{Queue_t^j, CPU_t^j, GPU_t^j, Bandwidth_t^j\}, j = \{1, \ldots, M\}$, where $Queue_t^j$ is the current unfinished task queue, $CPU_t^j$ and $GPU_t^j$ are the CPU and GPU computing powers (unit: TOPS), $Bandwidth_t^j$ is the bandwidth of $E_j$ (unit: Kbps). According to the current state $s_t$, the decision agent performs the action $a_t = \{a_t^i\}, i = \{1, \ldots, N\}$ for all drone. The action for drone $D_i$ is denoted as $a_t^i = \{E_j, T_k\}$, where $E_j$ is the edge connected to $D_i$, and $T_k$ is the feature analysis task performed on $E_j$ for the images of $D_i$. After taking action $a_t$, the decision agent receives feedback reward $r_t$ from the environment. We expect that the scheduled $T_k$ for $D_i$ could bring high accuracy, and the $E_j$ could real-time execute the task. Therefore, the reward $r_t$ is designed as:

$$r_t = Acc(output_t, label_t) - Time_t^e, \qquad (6)$$

where $output_t$ is the output result of $Task_t$, $label_t$ is the ground truth. $Acc(\cdot)$ is the accuracy score of $Task_t$, which is 1 if $output_t$ and $label_t$ are the same, otherwise is 0. $Time_t^e$ refers to the end-to-end latency of $Task_t$, consisting of computation, transmission, and queuing latency. The decision agent learns by maximizing the gain $G_t$, which is a function of the expected cumulative discount rewards:

$$G_t = E\left[\sum_{k=0}^{\infty} \gamma^k r_{t+k}\right], \qquad (7)$$

where $\gamma$ is the discount factor. Basis DQN uses an evaluation Q-network (parameterized by $\theta$), a target Q-network (parameterized by $\theta'$), and a trace pool to approximate the action value function. On this basis, model-assisted DQN uses the environment agent to assist in learning, which is trained by the following loss function:

$$L(\theta) = E\left[r_t + y_t - Q(s_t, a_t; \theta)\right], \qquad (8)$$

where $y_t$ is the target value of training. In model-assisted DQN, the target value $y_t$ is defined as:

$$y_t = \begin{cases} \gamma \max Q(s_{t+1}, a_{t+1}; \theta'), & if \text{ early training,} \\ E\left[\sum_{k=1}^{n} \gamma^k \left(r'_{t+k} \mid s, a\right)\right], & if \text{ late training,} \end{cases} \qquad (9)$$

where $r'$ represents the reward estimated by the environment agent. In the late training phase, the environment agent can output reliable estimations. Thus, $y_t$ could be estimated by the $n$ steps iterative output of the environment agent and the decision agent.

**Environment Agent Learning.** The goal of the environment agent is to learn the mapping from $(s_t, a_t)$ to $(s_{t+1}, r_t)$. The environment agent uses two Q-networks the same as the decision agent to respectively output the estimation of the next state $s'_{t+1}$ and the reward $r'_t$, which are trained by the MSE loss function. In the process of interaction between the decision agent and the environment, we obtain and store the real feedback data of the environment for training the environment agent. For offline training, we use all data stored in the trace pool for learning. During the online decision, only the Q-network that outputs $s'_{t+1}$ is fine-tuned by data from the previous period to prevent noise from contaminating the model.

### 4.3 Feature Analyzer

**Design.** We use a multi-feature anomaly detection algorithm as the feature analyzer [28], the workflow of which is as follows:

① **Feature Extract.** After receiving the images set from the drone, the edge device extracts the assigned feature of the images according to the scheduled result. Referring to the design in [28], we extract three types of features: i) *Velocity*: the movement speed of a person between adjacent frames, calculated based on optical flow [19]. ii) *Pose*: the key points of the human skeleton, generated by a posture detector [15]. iii) *Semantic*: the semantic information of the person images, obtained by the language-image encoder [27].

② **Density Estimate and Calibrate.** We estimate the density of extracted features to evaluate the degree of the anomaly, where a low estimated density indicates the anomaly. Each feature will be fitted with a separate estimator by $k$-NN to calculate the density and generate anomaly scores for the test samples. Moreover, we use min-max normalization to calibrate the anomaly scores to the same range to perform multi-feature fusion.

③ **Result Fuse and Output.** After generating anomaly scores in all edge devices, the scores are sent to the leader edge. Since the scores are calibrated, the fusion score can be generated by addition. The fusion score is used to compare with the threshold, if it exceeds the threshold, the current frame is considered abnormal.

**Table 1: Existing crowd anomaly detection datasets.**

| Dataset | Anomaly | Crowded | Disaster | Mulit-view |
|---------|---------|---------|----------|------------|
| USCD Ped1 [22] | indirect | ✗ | ✗ | ✗ |
| USCD Ped2 [22] | indirect | ✗ | ✗ | ✗ |
| CUHK Avenue [20] | indirect | ✗ | ✗ | ✗ |
| ShanghaiTech [21] | indirect | ✗ | ✗ | ✗ |
| UMN [24] | indirect | ✓ | ✗ | ✗ |
| Crowd-11 [12] | indirect | ✓ | ✗ | ✗ |
| MED [26] | both | ✓ | ✗ | ✗ |
| UBnormal [2] | both | ✗ | ✗ | ✗ |
| **ArmyStampede** | both | ✓ | ✓ | ✓ |

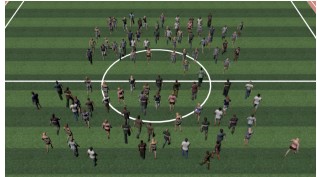 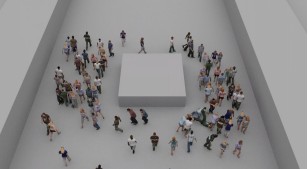

(a) Normal example.          (b) Abnormal example.

**Figure 7: Example of the ArmyStampede dataset.**

# 5 ARMYSTAMPEDE DATASET

## 5.1 The Definition of Crowd Disaster Anomaly

**Goal.** The causes of crowd disasters have been studied widely in sociology but are not in the realm of computer science. Thus, we try to define the anomaly that leads to crowd disasters from the view of sociology, which helps us to analyze the difference between the existing datasets and the dataset that we need.

**Definition.** Taking the New Year's stampede incident on Shanghai's Bund as an example, the evolution of crowd disaster can be summarized into three stages in sociology [11]: i) High density of the crowd. ii) Inconsistent flow direction. iii) Tumbling.

According to this evolution, we classify crowd disaster anomalies into two types: i) *Direct anomaly* that directly causes crowd disasters, like tumbling and collisions. ii) *Indirect anomaly* that increases the risk of crowd disaster but does not directly cause the disasters, such as high crowd densities and intersecting pedestrian flows.

## 5.2 Proposed Dataset

**Expected Dataset & Existing Limitation.** According to the above definition, we expect the dataset for detecting crowd disasters to have the following characteristics: i) Large-scale crowds. ii) Two types of anomaly. iii) Causing crowd disasters. iv) Multi-drone views shooting. However, to the best of our knowledge, there is no dataset available for our expectations. Table 1 presents statistics about the most commonly utilized datasets in CAD, which are limited by the following factors: i) Anomaly is limited to indirect type which won't cause crowd disasters, like people crossing the road. ii) The size of the crowd in the scene is small, usually less than 10 people. iii) Most datasets are only based on a single fixed-camera view.

**Goal & Problem.** Due to the existing limitations, our goal is to collect a dataset of crowd disasters to evaluate the performance of Air-CAD. The collection of actual crowd disaster data, however, poses major challenges and needs to ensure ethical standards. Therefore, an alternative way is to create a simulated dataset.

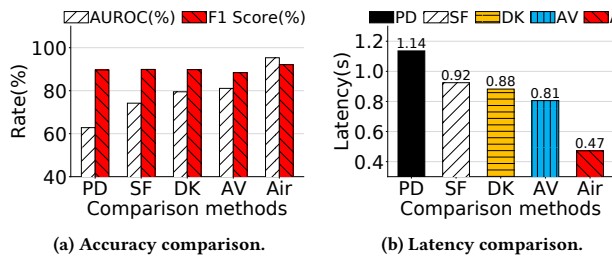

(a) Accuracy comparison.          (b) Latency comparison.

**Figure 8: Overall performance comparison.**

**Design.** We generated a dataset for detecting crowd disasters, called ArmyStampede. To achieve our goals, we design the dataset by the following steps: i) *Individual agent modeling*. Our crowd simulation generates collective crowd behaviors by modeling individual agents on three levels: personality [16], emotions [13], and behavior. ii) *Disaster event design.* A collision detection logic is designed for the agents to simulate the process of crowd stampede disasters. iii) *Crowd simulation and recording*. We use the most advanced 3D crowd simulator [1] to conduct simulations, which allows us to freely model scenes, place agents, and record from drone views.

**Dataset Summary.** Our ArmyStampede dataset contains 72000 Full HD frames. Table 1 compares ArmyStampede against others, the main characteristics of which include: i) *Large-scale crowd*: we simulate different numbers of crowds, from 20 to 100. ii) *Crowd disaster anomaly*: our dataset has two types of anomaly that can cause crowd disasters. The indirect anomaly includes high crowd density, conflict in crowd flow, and panic in the crowd. The direct anomaly includes collisions between individuals, tripping, and stampeding. iii) *Multi-view recording*: we shoot the crowd from different angles and distances to simulate the drone views with different shooting parameters. Specifically, the shooting distances are from 5 to 25 meters, and the shooting angles are from 0 to 90 degrees.

# 6 EVALUATION

## 6.1 Hardware Implementation and Dataset

**Hardware.** For the drones, we use an NVIDIA Jetson Xavier NX as the onboard computing device. For the edge devices, we use a PC that has an Intel Xeno E5-2687W v4 CPU and an NVIDIA RTX 2080Ti GPU with 12 GB of memory.

**Implementation.** We deploy Air-CAD on three drones and three edge devices to demonstrate Air-CAD's operation. We implement the zoom detector on drones and the scheduler on the leader edge using PyTorch 1.10. We deploy feature analyzers on edge devices according to [28]. Data transfer between devices via 2.4GHz WLAN at up to 18 Mbps upload/download rate.

**Data.** We use our new dataset to evaluate the performance of algorithms under various shooting parameters, which includes crowd-disaster images recorded from different drone views.

## 6.2 Overall Performance

We first evaluate the overall performance of Air-CAD. At a shooting instant (0.15 seconds), each drone captures a frame of the crowd and then performs the CAD pipeline. We define end-to-end CAD as a *task* consisting of person detection and feature analysis.

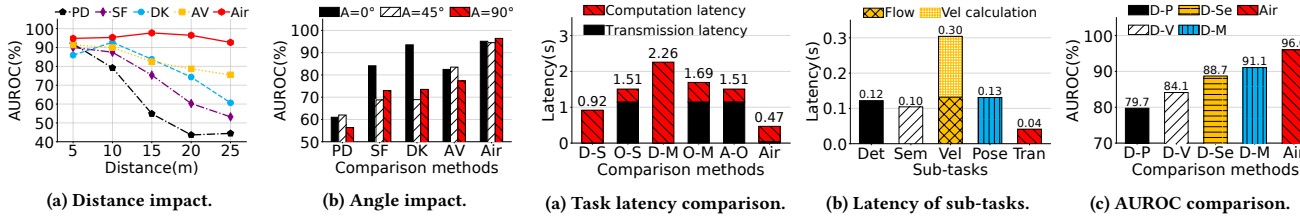

**Figure 9: Impact of shooting parameters.**

**Figure 10: Framework performance evaluation.**

**Baseline Methods.** We select four mainstream CAD algorithms implemented in the standard anomaly detection library [3] for comparison, including the following algorithms: i) *PaDiM* (PD) [8]. ii) *STFPM* (SF) [32]. iii) *DFKDE* (DK). iv) *AI-VAD* (AV) [28].

**Accuracy.** Figure 8a shows the overall accuracy comparison between Air-CAD and the baselines. Air-CAD has the highest AUROC of 95.33% and F1 Score of 92.11%. Due to the designed person detector and feature scheduler for multiple drones, Air-CAD shows a significant 14.25% improvement in AUROC over AI-VAD (the best baseline), which also utilizes the same feature analysis methods.

**Task Latency.** Figure 8b shows that Air-CAD achieves the lowest latency of only 0.47 seconds, which is 1.72 times faster than the optimal baseline (AI-VAD). This is attributed to Air-CAD's fast person detection and parallel execution across multiple devices.

**Shooting Parameters Impact.** Figure 9 depicts the performance comparison between Air-CAD and the baselines under different shooting parameters. As shown in Figure 9a, Air-CAD maintains 92.69% AUROC at 25 meters distance, while the AUROC of baselines decay in varying degrees. As shown in Figure 9b, Air-CAD delivers consistently high AUROC of more than 90% at all angles.

## 6.3 Air-CAD's Framework Performance

We deploy the Air-CAD pipeline on baseline frameworks to analyze the impact of the frameworks on accuracy and latency.

**Baseline Framework.** We compare several baseline frameworks for drone vision analysis, mainly as follows:

① **Single Drone Onboard Analysis Single Feature (D-S).** This framework uses a single drone, analyzing only one specific feature. D-P, D-V, and D-Se represent the analysis of the pose, velocity, or semantics, respectively.

② **Single Drone Offloading Analysis Single Feature (O-S).** Compared to D-S, this framework offloads the complete image to the edge for person detection and single-feature analysis.

③ **Single Drone Onboard Analysis Multiple Feature (D-M).** Compared to D-S, this framework uses a single drone for onboard computation, analyzing all three types of features.

④ **Single Drone Offloading Analysis Multiple Feature (O-M).** Compared to D-M, this framework offloads the complete image to the edge for person detection and the analysis of all three features.

⑤ **Air-CAD Complete Offloading (A-O).** Compared to the complete Air-CAD, A-O offloads the complete image to edge devices instead of performing person detection on drones.

**Task Latency.** Figure 10a shows the comparison of task latency among different frameworks. The task latency of the complete Air-CAD is much smaller than other frameworks, up to 0.47s. Air-CAD is 1.95 times faster than D-S and 4.81 times faster than D-M. Due to

the limited onboard computing resources of drones, D-S and D-M experience high computation latency of 0.92s and 2.26s. Air-CAD is 3.21 times faster than O-S, 3.59 times faster than O-M, and 3.21 times faster than A-O. This is due to their huge transmission latency caused by transmitting Full HD images, which reaches 1.14s.

We further analyze the cost of each sub-task (i.e., person detection, three types of feature analysis, and transmission) in Air-CAD's pipeline. As shown in Figure 10b, different feature analysis tasks have varying computation latency, which can be effectively distributed among suitable edge devices by Air-CAD's scheduler.

**AUROC.** Figure 10c shows the AUROC comparison of different frameworks. The complete Air-CAD achieves the highest AUROC of 95.33%. The AUROC of analyzing a single feature is low because it is difficult to characterize two types of crowd disaster anomalies by using only one type of feature. Analyzing multiple features on a single drone view can enhance accuracy but is limited. Air-CAD could analyze the multi-features from multi-drone views with various shooting parameters to achieve satisfactory performance.

## 6.4 Air-CAD's Module Performance

**Air-CAD's detection performance.** We compare the zoom detector with the mainstream detectors: Yolov5-s, Yolov5-l, and TPH-Yolov5. Figure 11a shows that the zoom detector has the highest accuracy compared to baseline detectors. Figure 11b shows that the zoom detector has better detection cost-effectiveness than the baseline detectors, which can complete the detection within 43 milliseconds and the accuracy is 86.16% at 25 meters shooting distance.

**Air-CAD's schedule performance.** We compare the batch latency and devices' queue length of Air-CAD with and without the scheduler when processing different numbers of tasks. As shown in Figure 12a, the batch latency of Air-CAD with the scheduler is at least about 22.6% lower than that of Air-CAD without the scheduler, which has high queue latency. As shown in Figure 12b, without the scheduler, the feature analysis with high computation latency will be continuously assigned to the same device, resulting in a growing task queue length. The scheduler can manage the workload by assigning feature analysis tasks to suitable edge devices.

## 7 REAL-WORLD FLIGHT

## 7.1 Implementation and Real-flight Experiment

**Drone & Edge Device.** We use three Q250 quadcopters for drones in the experiment. Each drone is equipped with a PIXHAWK 4 Mini flight control, an M8N GPS, a 4K action camera, and an NVIDIA Jetson Xavier NX. We use three edge devices that are implemented the same as the implementation in §6.

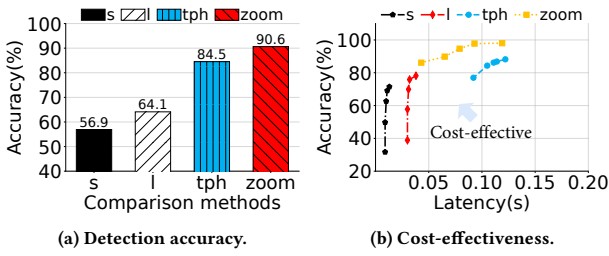

(a) Detection accuracy.   (b) Cost-effectiveness.

**Figure 11: Detection performance evaluation.**

**Table 2: Real-flight evaluation results.**

| Experiment | AUROC | F1 score | Average latency |
|------------|-------|----------|-----------------|
| Total      | 94.86% | 87.88%  | 0.65s           |
| Dis 10m    | 99.39% | 98.39%  | 0.71s           |
| Dis 20m    | 95.22% | 83.45%  | 0.66s           |
| Dis 30m    | 89.52% | 81.80%  | 0.59s           |

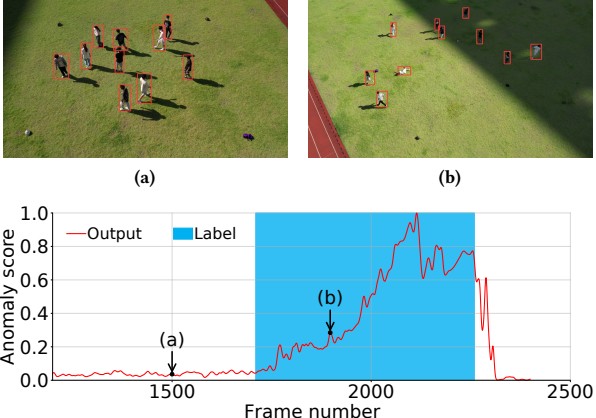

(a)                    (b)

**Figure 14: Real-world flight visualization on anomaly score, where: (a) normal frame at 10m shooting distance and 0° shooting angle; (b) abnormal frame at 20m shooting distance and 45° shooting angle.**

**Real-flight Experiment.** We deploy and evaluate Air-CAD on drones and edge devices in real-world experiments. The experiments simulate a crowd anomaly with ten volunteers in an outdoor scene. Each experiment contains a normal test and an abnormal test. The normal test involves people moving smoothly, whereas the abnormal test involves people tumbling and the crowd scattering. We totally run three experiments by varying the shooting distances (10m, 20m, 30m) of the drones. In each experiment, three drones that capture Full HD resolution images are located at three shooting angles (0°, 45°, 90°).

## 7.2 Real-flight Evaluation Results

Table 2 shows that Air-CAD achieves high accuracy in real-world evaluation. As shooting distances increase, Air-CAD can still maintain high accuracy. Figure 14 shows the visualization of the output anomaly scores with the corresponding frames at that moment. Air-CAD can detect the occurrence of the anomaly in the crowd. In normal frames, as shown in Figure 13a, the anomaly scores remain

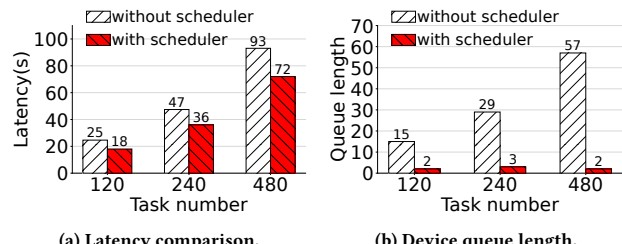

(a) Latency comparison.   (b) Device queue length.

**Figure 12: Scheduling performance evaluation.**

at a low level. When the crowd starts to flee, as shown in Figure 13b, the anomaly score starts to rise sharply.

## 8 RELATED WORK

**Drone-view Person Detection.** Recently, detection techniques have been applied to drones to detect people, which face a variety of challenges because of drones flying at high altitudes. Chen et al. propose an RRNet that combines an anchor-free detector with a re-identification module to achieve high accuracy in drone-view detection [6]. Zhu et al. propose TPH-Yolov5 that augments YOLOv5 with a TPH head to detect objects of different scales and integrates the CBAM to find attention areas [37]. Deng et al. achieve an end-to-end drone view detection by a global-local detection network, a simple yet efficient self-adaptive region selecting algorithm, and a super-resolution network [9]. In addition, insufficient computing resources hinder the real-time execution of complex DNNs on drones, which can be alleviated by dynamic DNNs. Surat et al. propose a framework that could exit the DNN early [31]. Wang et al. design a modified residual network that uses a gate to skip convolutional blocks based on the activations of the previous layer [33].

**Crowd Anomaly Detection (CAD) on Drones.** Recently, images from drones have been used to detect crowd anomalies. Jordan et al. use a CNN and an RNN to detect crowd anomaly, which are trained with drone-view images [17]. Rezaee et al. propose a real-time monitoring strategy based on deep transfer learning and drone internet for detecting abnormal behavior in crowds [29]. Danilo et al. propose a single-class support vector machine (OC-SVM) anomaly detector based on customized Haralick texture features for low-altitude aerial video surveillance [5]. CAD on real drones has not yet been extensively tested due to restricted flight conditions.

## 9 CONCLUSION

In this paper, we propose Air-CAD, an edge-assisted multi-drone network for crowd anomaly detection under real flight conditions. To achieve fast and accurate person detection, we design a zoom detector for Air-CAD to dynamically adjust the depth of model inference and focus on key channels of the feature map based on the shooting distances. A feature scheduler in Air-CAD determines which tasks and edge devices are best suited for drones according to their shooting parameters and the computing/networking resources of their edge devices. Moreover, we generate a crowd disaster dataset, which is the first dataset for CAD recorded from multiple drone views. Our simulation and real-world experiments show that Air-CAD can achieve fast and accurate CAD on real drones, providing security for the smart city in the web era.

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

# A    DETAILED DESIGN OF PROPOSED DATASET

**Table 3: Correlation between Emotions, PAD Values, and Behaviors of the individual agent.**

| Emotions | PAD Values | Behaviors |
|---|---|---|
| Happy | P>0 | Standing, Jogging, Talking |
| Sad | P<0, A<0 | Walking alone |
| Angry | P<0, A>0, D>0 | Chasing, Pushing |
| Fearful | P<0, A>0, D<0 | Fleeing, Dodging, Crawling |

**Individual agent modeling.** We mainly model individual agents in the crowd through three levels:

i) *Personality modeling.* We assign unique personalities to each individual agent, making them have different behavioral norms. We use the PEN model [14] to represent personality by dividing it into three categories: Psychoticism, Extraversion, and Neuroticism. The three categories of personality correspond to three kinds of behavioral norms, provided in [16]. We randomly initialize individual agents' personalities, so that to achieve a vivid crowd simulation.

ii) *Emotion modeling.* During the process of crowd simulation, each agent has emotions that can influence the behaviors. The emotions are quantified by the PAD values [23], which uses three orthogonal components to represent emotions: Pleasure, Arousal, and Dominance. Table 3 shows the specific relationship between emotions and PAD values. Emotions can be influenced by the behaviors of other agents [13], manifested as the change in PAD values. Specifically, if another agent within the range of the current agent's perception is engaged in a specific behavior, the PAD value of the current agent will linearly shift towards the direction of the PAD value corresponding to the behavior.

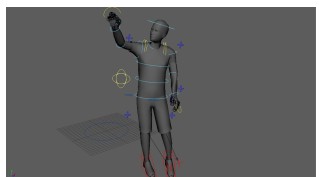 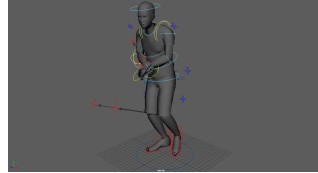

| (a) Talking with happy emotion. | (b) Dodging with fearful emotion. |
|---|---|

**Figure 15: Examples of behaviors under different emotions.**

iii) *Behavior modeling.* As the emotion changes, the agent's behavior changes as well. We design different behavior sets for different emotions, shown in Table 3. Behaviors based on emotions are diverse and logical, enabling a realistic simulation of the crowd.

# B    ABLATION EXPERIMENT OF FLIGHT CONDITIONS AWARENESS

We analyze the impact of perceiving flight conditions on performance through ablation experiments, including shooting distances, shooting angles, and edge device status. Specifically, we set the input of flight conditions to zero to simulate situations in which the Air-CAD is not aware of flight conditions.

**Distance.** Figure 16a illustrates how the detector performs without distance perception. A large shooting distance does not greatly affect the performance, while a small shooting distance significantly reduces it. We found that when distance is not perceived, the detector tends to stop early (i.e., infer fewer blocks) to achieve low inference loss, leads to low accuracy at small shooting distances.

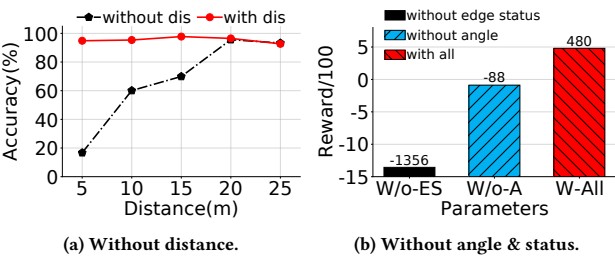

| (a) Without distance. | (b) Without angle & status. |
|---|---|

**Figure 16: Impact of w/o flight conditions input.**

**Angle & Edge Status.** Figure 16b shows the change of the scheduler's reward when the shooting angle or edge status is not perceived. The rewards given for not perceiving the angle and the device status are 1.168 times and 3.825 times smaller than the reward when they are perceived. Despite not perceiving the shooting angle, the scheduler can still assign the appropriate edge devices to different feature analysis tasks, but could not assign the appropriate feature analysis task to the drones. When not perceiving the device status, the scheduler could not assign appropriate edge devices for feature tasks, resulting in a heavy workload on the system.

# C    PARAMETERS OF ZOOM DETECTOR.

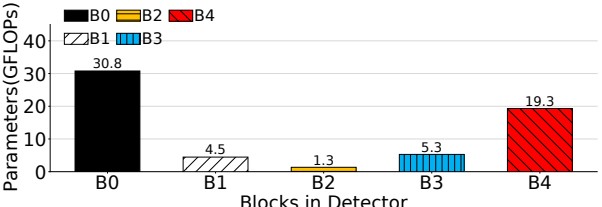

**Figure 17: Parameters of Zoom Detector.**

Figure 17 shows the computational parameters of different blocks in the zoom detector. B0 is the backbone that has the most parameters to ensure the extraction of valid information from the image. B1 to B4 are blocks that can output results from shallow to deep. Deep blocks have a high number of parameters to extract high-level information, which is unnecessary at far shooting distances. The zoom detector can select the appropriate block for output based on the shooting distance, to achieve high performance with less computational parameters.

# D    DEPLOYMENT RECOMMENDATIONS FOR REAL-WORLD FLIGHT

Air-CAD demonstrates its feasibility in real-flight evaluation. In light of the results, we provide readers with recommendations for better deployment of Air-CAD. To monitor crowds in outdoor scenarios, Air-CAD requires 3 drones to achieve 90% AUROC. Drones should capture images from different shooting angles to enhance the performance of feature analysis, from 0° to 90°. The shooting distances of the drones can vary, ranging from 10m to 30m. The flight speed of the drone should be kept below 1m/s to reduce image blur. WLAN is recommended for communication, which has greater bandwidth for multi-device connection compared with LTE.