# OpenReview forum: "Air-CAD: Edge-Assisted Multi-Drone Network for Real-time Crowd Anomaly Detection"
_ACM.org/TheWebConf/2024/Conference — TheWebConf24 Oral_

### Official Review · Reviewer_9ZFj · 2023-10-30

**Novelty:** 6
**Technical Quality:** 7

**Review:**

The paper presents, Air-CAD, an edge-assisted multi-drone network for crowd anomaly detection (CAD). After investigating the impact of flight conditions on CAD performance as motivation, it presents a design Air-CAD consisting of person detection and multi-feature analysis. For accurate and fast detection, the design includes primarily three modules, 1) Zoom Detector to dynamically adjust the depth and focus based on the drones’ shooting distances for fast and accurate person detection, 2)  Feature Scheduler to efficiently offload data and feature analysis tasks to suitable edge devices, and 3) Feature Analyser that is a multi-feature anomaly detection algorithm. The system is comprehensively evaluated the system using both simulation and real-world experiments. The simulation relies on a newly collected dataset, i.e., ArmyStampede, that is synthetic yet large-scale and comprehensive, encompassing direct and indirect crowd anomalies. The proposed system outperforms the selected benchmarks, providing fast (lowest latency) and accurate CAD (highest AUROC).

**Pros:**
- The paper is well-written and organized, with a logical flow of ideas.
- The system design and methodology are sound.
- To inspire new architectural stages that improve performance, the authors investigated the performance of general CAD and derived insights for choosing impacting parameters.
- The authors introduce a novel approach by combining drone networks with edge computing, supported with novel modules.
- The evaluation is comprehensive with suitable metrics, figures, and tables that enhance the clarity of the findings.

**Cons:**
- While I believe the work is solid and original, it doesn’t seem very suitable for submission to the WebConf conference.
- AUROC has inherent limitations in imbalanced datasets, with significant disparity between the number of normal and abnormal instances. AUROC can still be high even if the model's performance on the minority class is poor. In such cases, AUROC is not enough, and other metrics such as a precision-recall curve or F1 score, would convey a clear picture of the performance.
- Figure 8 shows that Air-CAD performs similarly or slightly better, compared to benchmarks in terms of F1 score. This indicates that the dataset is imbalanced.
- Some writing issues led to confusion. For instance, the texts refer to Figure 13a and Figure  13b, while Figure 13 is entirely missing. Perhaps 14a & 14b instead! In Figure 4, there is a typo. Perhaps  “Shooting Parameters” not “Paraments”.

**Questions:**

- “This track solicits novel research contributions describing the construction of systems architecture, and performance related to the Web, and Web-based mobile and ubiquitous computing”. What makes this work relevant to this WebConf track? The authors mentioned WoT in the intro. However, it is not clear how Air-CAD could be integrated into WoT.
- In Figure 2a, you use AUROC to study the impact of shooting parameters on overall AUROC, while you use accuracy to study the impact of shooting parameters on detection accuracy in Figure 3a. What is the difference and how did you measure the accuracy in Figure 3a? How did you measure accuracy in Figure 11?
- What is the percentage of normal, and abnormal instances, or direct and indirect anomalies in the ArmyStampede dataset, and how it is labeled?
- Figure 13 is missing. I believe you mean 14a & 14b.
- In Figure 4, there is a typo. I believe you mean “Shooting Parameters” not “Paraments”

**Ethics Review Description:**

No ethical issues

**Reviewer Confidence:**

4: The reviewer is certain that the evaluation is correct and very familiar with the relevant literature

**Scope:**

2: The connection to the Web is incidental, e.g., use of Web data or API

---

### Official Review · Reviewer_YYk3 · 2023-11-22

**Novelty:** 5
**Technical Quality:** 5

**Review:**

pros:
1.The Air-CAD system has the advantages of high efficiency and real-time. Through air-ground coordination, dynamic adjustment of drone shooting distance and angle, and deployment of edge devices, it achieves high accuracy, rapidity and real-time crowd anomaly detection. At the same time, Air-CAD has high practical value in practical applications.
2.Air-CAD proposes a new dataset（called ArmyStampede）to simulate human panic escape, which is derived from the recording of various drone perspectives, and provides a new experimental verification method for drone crowd anomaly detection.
3.The experiment of the paper is very sufficient.The experimental results show that Air-CAD performs well in both simulated and real environments, achieving 95.33 % AUROC and real-time inference delay within 0.47 seconds.
cons:
1.In this paper, the security and privacy protection measures of drone network in data transmission, storage and processing are not mentioned. In practical applications, these problems need to be paid close attention to and solved.
2.The paper does not discuss the feasibility and sustainability of Air-CAD in practical applications, such as drone battery life and drone control difficulty.
3.The paper does not fully discuss the applicability of Air-CAD in different scenarios, such as different scale activities, outdoor and indoor environments.

**Questions:**

1.During the practical application of the Air-CAD system, it is essential to implement security and privacy protection measures in data transmission, storage, and processing to safeguard against potential risks and maintain user privacy.
2.The paper has the following deficiencies in discussing the feasibility and sustainability of the Air-CAD system :
	2.1 Drone battery life: The paper does not discuss how to ensure the battery life of drones during long-term operation.
	2.2 Drone control difficulty: The paper does not address how to solve the problem of controlling multiple drones, thus improving operation efficiency and reducing labor costs.
	2.3 Cost-effectiveness: The paper does not discuss the cost-effectiveness of the Air-CAD system, including drone, edge device, and operational 	costs.
	2.4 Regulation and policy:Real-world testing of CAD on drones has been limited due to constraints on flight conditions.
3.The paper fails to comprehensively explore the applicability of Air-CAD in various scenarios. For instance, it does not thoroughly discuss how well the system performs in different scale activities. Additionally, the paper does not provide sufficient information on how Air-CAD adapts to outdoor and indoor environments.

**Reviewer Confidence:**

2: The reviewer is willing to defend the evaluation, but it is likely that the reviewer did not understand parts of the paper

**Scope:**

3: The work is somewhat relevant to the Web and to the track, and is of narrow interest to a sub-community

---

### Official Review · Reviewer_zdPM · 2023-11-24

**Novelty:** 3
**Technical Quality:** 4

**Review:**

This paper proposes an Air-CAD that is an edge-assisted multi-drone network for crowd anomaly detection. It achieves high accuracy and real-time inference latency. However, the idea of this paper is not novel enough, and it lacks comparisons with state-of-the-art methods.
Besides, the authors used the wrong template for the WWW paper.

**Questions:**

1. Use the wrong template.
2. in the submitted paper, the authors use model-assisted DQN for the scheduler. Why DQN is selected? Could you clarify the reason why the DQN was selected?
3. Besides, could you compare the proposed model-based DQN method with other deep reinforcement learning methods, such as PPO, A3C, etc.?
4. Although the authors have evaluated the performance of the proposed framework in terms of different aspects, no details on the parameter setting are provided in the evaluation section. Could you please provide more details on the evaluation of the proposed Air-CAD framework, such as parameter setting, and datasets, etc.?
5. In the Ablation experiment of flight conditions awareness, is it reasonable to set the input of flight conditions to zero as the Air-CAD as unaware of the flight conditions?

**Reviewer Confidence:**

3: The reviewer is confident but not certain that the evaluation is correct

**Scope:**

3: The work is somewhat relevant to the Web and to the track, and is of narrow interest to a sub-community

---

### Official Review · Reviewer_qs8i · 2023-11-26

**Novelty:** 2
**Technical Quality:** 6

**Review:**

This paper proposes a design and application of a edge-network with drones for real-time crowd disaster anomaly detection. The paper focuses on a "zoom detector" for person detection using image processing and "feature scheduler" for anomaly detection. The paper includes evaluation from a 3D simulator and a real-world data collected with a limited setup.

Pros:
- The paper includes both simulated and real-world experiment for crowd anomaly detection.
- The paper considers various engineering technique in their design of the drone-network.

Cons:
- The application seesm a bit not well motivated. It seems rather an over-engineered solution for the crowd anomaly detection.
- The solution may have problem with practical application aspects.
- The work does not seem relevant to the Web.

Detailed comments below:
- The setup of the real-world experiment seems very different than the simulated setup. I am not sure if any crowd disaster could be detected with only 10 participant in a relatively large ground. Real crowd disasters may happen in very different urban setups with buildings or unexpected environments.
- The three scenarios seem rather arbitrary although it is taken from a study. Crowd behaviors may be more complex and detection of them might not be as straight-forward.
- Related work seems rather unsatisfactory. The work related to crowd behavior detection is not only based on using drones. Actually, the drone application seem more on the exotic side compared to existing work on crowd behaviors detection.
- Although mentioned in the title, abstract and introdcution, the paper does not really focus on the networking problems, it is rather more focusing on the machine learning computation through image processing.
- The motivation is a bit not so clear, considering the given background and related work. For instance, it is not clear why a drone-edge-network is needed for a solution, whereas many cameras are already deployed on the ground (with their possible edge capabilities).It is also not clear how a real solution can be implemented considering existing problems of drones (e.g., battery) and spontaneity of crowd gatherings.
- The solution seems to have practical aspects such as maintaining such an edge network and drones and operating in a real setup.
- In addition to the practical application, the ethical concerns would still apply (even though mentioned in the paper). This is  mentioned in the paper even for data collection in research purpose. It is hard to imagine when they will be solved and, when they are solved, if the current engineering solutions (e.g., processing capability) would still be relevant.

**Questions:**

Although the authors listed various considerations on the engineering solution, I would like to ask a few question regarding the real application of the solution.

- What would be a real application scenario for such a solution?
- What would be the cost of an edge-network in a real setup and how it can be maintained?
- How could the ethical aspects be addressed for the deployment of the solution?

**Reviewer Confidence:**

3: The reviewer is confident but not certain that the evaluation is correct

**Scope:**

1: The work is irrelevant to the Web

---

### Official Review · Reviewer_g4cd · 2023-11-28

**Novelty:** 5
**Technical Quality:** 5

**Review:**

Overall, I think this is an interesting work. It reveals interesting observations in the motivational studies, and design corresponding modules based on these observations. The evaluation is based a generated dataset and a real-world setting. The paper is also well presented.

**Questions:**

What is the difference between generally person detection and crowd anomaly? What new challenges can this objective bring to the problem?

The reviewer also suggests the authors clearly state the limitations of others in the related work section.

**Reviewer Confidence:**

2: The reviewer is willing to defend the evaluation, but it is likely that the reviewer did not understand parts of the paper

**Scope:**

3: The work is somewhat relevant to the Web and to the track, and is of narrow interest to a sub-community

---

### Decision · Program_Chairs · 2024-01-22

**Decision:**

Accept (Oral)

**Comment:**

Overall, the paper presents a research work on an edge-assisted multi-drone network for crowd anomaly detection. I summarise the pros and cons from the reviewers as follows.

 Pros:
 Interesting observations and corresponding design modules based on those observations.
 Evaluation based on both generated dataset and real-world setting.
 Well-presented paper.
 Includes both simulated and real-world experiments.
 Considers various engineering techniques in the design.
 Achieves high efficiency and real-time crowd anomaly detection.
 New dataset for experimental verification.
 Sufficient experiments with good results.
 Well-written and organized paper.
 Sound system design and methodology.
 Novel approach with drone networks and edge computing.
 Comprehensive evaluation with suitable metrics.

 Cons:
 Lack of clear limitations of related work.
 Application seems over-engineered for crowd anomaly detection.
 Potential issues with practical application.
 Not relevant to the Web.
 Lack of novelty and comparisons with state-of-the-art methods.
 Use of wrong template.
 Missing details on parameter settings and datasets.
 Questionable input setting in the ablation experiment.
 No discussion on security and privacy protection measures.
 No discussion on feasibility and sustainability in practical applications.
 No discussion on applicability in different scenarios.
 Not suitable for submission to the WebConf conference.
 Limitations of AUROC in imbalanced datasets.
 Confusion in figure references and typos in the text.

 The paper has a balanced pros and cons. There is an issue about the relevant to the web conference pointed out by the reviewers.